# Nature Nano-Barrier: HPMC/MD-Based *Lactobacillus plantarum* Pickering Emulsion to Extend Cherry Tomato Shelf Life

**DOI:** 10.3390/foods14152729

**Published:** 2025-08-05

**Authors:** Youwei Yu, Tian Li, Shengwang Li, Silong Jia, Xinyu Yang, Yaxuan Cui, Hui Ma, Shuaishuai Yan, Shaoying Zhang

**Affiliations:** College of Food Science, Shanxi Normal University, Taiyuan 030031, China; 223420028@sxnu.edu.cn (T.L.); liswang98@163.com (S.L.); 223420006@sxnu.edu.cn (S.J.); 223420014@sxnu.edu.cn (X.Y.); 224420002@sxnu.edu.cn (Y.C.); huima@sxnu.edu.cn (H.M.); yanshuaidouble98@163.com (S.Y.)

**Keywords:** Cherry tomato, Coating preservation, HPMC, *Lactobacillus plantarum*, MD, W/W Pickering emulsion

## Abstract

To improve the postharvest preservation of cherry tomatoes and combat pathogenic, both bacterial and fungal contamination (particularly *Alternaria alternata*), a novel biodegradable coating was developed based on a water-in-water (W/W) Pickering emulsion system. The emulsion was stabilized by *L. plantarum* (*Lactobacillus plantarum*), with maltodextrin (MD) as the dispersed phase and hydroxypropyl methylcellulose (HPMC) as the continuous phase. Characterization of emulsions at varying concentrations revealed that the optimized W/W-PL^8 film exhibited superior stability, smooth morphology, and low water vapor permeability (WVP = 220.437 g/(m^2^·24 h)), making it a promising candidate for fruit and vegetable preservation. Furthermore, the coating demonstrated strong antioxidant activity, with scavenging rates of 58.99% (ABTS) and 94.23% (DPPH), along with potent antimicrobial effects, showing inhibition rates of 12.8% against *Escherichia coli* and 23.7% against *Staphylococcus aureus*. Applied to cherry tomatoes, the W/W-PL^8 coating significantly reduced respiration rates, minimized decay incidence, and maintained nutritional quality during storage. Remarkably, the coating successfully controlled *Alternaria alternata* contamination, enhancing the storage duration of cherry tomatoes. These findings highlight the potential of W/W-PL^8 as an eco-friendly and functional packaging material for fresh produce preservation.

## 1. Introduction

Maintaining a balanced diet requires regular consumption of fresh fruits and vegetables, which are essential for human health. Cherry tomato (*Solanum lycopersicum* var. *cerasiforme*), a widely cultivated crop globally and an important commercial fruit in China’s agricultural economy, is rich in antioxidant compounds such as lycopene and vitamin C (Vc). These components effectively scavenge free radicals and reduce the risk of disease [1]. However, fruits and vegetables are highly susceptible to plant pathogens during cultivation, postharvest transportation, and storage, leading to spoilage [2]. Among these pathogens, *Alternaria alternata* (Nees: Fr) is a predominant causative agent. The necrotic diseases it induces not only reduce crop yields but also contribute to postharvest decay, resulting in significant economic losses [3]. Furthermore, a cherry tomato is a climacteric fruit characterized by continuous and active respiratory metabolism during ripening. This metabolic activity accelerates the transition from ripening to spoilage under the combined stress of interrupted nutrient supply and pathogen invasion [4].

At present, although traditional food packaging materials have cost advantages, their non-degradable characteristics can easily cause “white pollution”. In view of the key role of packaging materials in delaying food quality deterioration, the development of biodegradable natural materials (such as proteins, polysaccharides, and lipids) has become a research focus [5]. Hydroxypropyl methylcellulose (HPMC), as a cellulose derivative, is endowed with multiple functional properties by the introduction of hydroxypropyl and methyl groups. This water-soluble biopolymer is often used as an emulsifier, stabilizer, and film-forming agent in the food industry [6], and its excellent biocompatibility has also made it a research hotspot in the field of bio-based films [7]. It is worth noting that although HPMC can play a fresh-keeping role by regulating fruit surface characteristics [8], it lacks antibacterial activity and needs to be combined with antibacterial components to improve comprehensive performance. Another polysaccharide material, maltodextrin (MD), is often used for encapsulation and thickening of bioactive substances because of its low cost and effective blocking of oxygen [9]. HPMC exhibits excellent film-forming and barrier properties, while MD serves as a carbon source for *L. plantarum*, sustaining probiotic activity to enable continuous release of antibacterial metabolites. The hydroxyl groups of HPMC form intermolecular hydrogen bonds with MD’s glycosidic bonds, enhancing film density and demonstrating superior synergy.

Compared with chemical antimicrobial agents, inducing postharvest disease resistance in fruits and vegetables using antagonistic microorganisms and their metabolites is more ecologically sustainable. Luo et al. demonstrated that such biological control methods can effectively suppress postharvest diseases [10], with commonly utilized microorganisms including bacteria, molds, and yeasts [11]. Taking lactic acid bacteria (LAB) as an example, these Gram-positive bacteria can produce organic acids (such as lactic acid and citric acid), hydrogen peroxide, and bacteriocins through metabolic processes, exhibiting significant antagonistic effects against various spoilage bacteria and fungal pathogens (*Alternaria alternata*). Specifically, LAB-derived organic acids disrupt fungal cell membrane integrity, while bacteriocins inhibit spore germination of *A. alternata* [12]. When LAB are incorporated into Pickering emulsions and applied as coatings on the surface of fruits and vegetables, they can inhibit the proliferation of pathogenic microorganisms through the sustained release of antimicrobial metabolites, thereby extending the freshness period [13].

Water-in-water Pickering emulsion (W/W-PE) is a system formed by mixing two immiscible polymers in an aqueous solution, resulting in the separation of two distinct phases [14]. The stabilizers used must be food-grade particles to ensure safety, and probiotics are ideal candidates due to their stability and inherent antimicrobial properties. In this study, *L. plantarum* Y-35 was innovatively employed as both a solid-phase stabilizer for the Pickering emulsion and a continuous antimicrobial agent, leveraging its inherent surface activity and bacteriocin production capabilities. By enhancing the adhesion of the emulsion to the fruit surface, the antimicrobial metabolites produced by bacterial metabolism effectively blocked external contamination, thereby maintaining the postharvest quality of cherry tomatoes.

This study has certain limitations. The incorporation of *Lactobacillus plantarum* may compromise mechanical strength and water vapor permeability (WVP) barrier properties compared to conventional polysaccharide coatings. Currently, few composite coating designs can simultaneously achieve high antimicrobial activity, excellent mechanical performance, and controlled degradability. Our research aims to overcome these limitations by developing a probiotic-polysaccharide composite system that addresses both microbial spoilage (particularly *A. alternata*) and physiological deterioration in fresh produce.

## 2. Materials and Methods

### 2.1. Materials

Cherry tomatoes (*Solanum lycopersicum* var. *cerasiforme*) of the Pink Baby variety were purchased from Huilong Market in Jinzhong City, Shanxi Province. Hydroxypropyl methylcellulose (HPMC, analytical grade) was obtained from Aladdin Reagents (Shanghai, China), and maltodextrin (food grade) was purchased from Shandong Sida Biotechnology Co., Ltd. (Weifang, China). *L. plantarum* Y-35 was isolated from naturally fermented corn silage (60 days) collected from a pasture silage cellar in Siziwang Banner, Inner Mongolia Autonomous Region, China. *Alternaria alternata* (BNCC 115062), *Escherichia coli* (ATCC 11229), and *Staphylococcus aureus* (CMCC 26003) were purchased from Beijing Nabio Biotechnology Co., Ltd. (Beijing, China). De Man, Rogosa, and Sharpe (MRS) broth was purchased from Haibo Biotechnology Co., Ltd. (Qingdao High-tech Industrial Park, Qingdao, China).

### 2.2. Culture and Treatment of Lactobacillus Y-35

The Y-35 strain was inoculated into 10 mL test tubes containing MRS broth and incubated for 24 h. The cultures were then centrifuged at 10,000× *g* (H1850R, Hunan Xiangyi Laboratory Instrument Development Co., Ltd., Changsha, China) for 20 min, after which the supernatant was discarded, and the precipitated bacterial biomass was collected for incorporation into the coating. To provide growth conditions, MRS broth (1.0%, *w*/*v*) was added to the W/W Pickering emulsion, yielding suspensions with concentrations of 1 × 10^10^, 1 × 10^8^, and 1 × 10^6^ (CFU) g^−1^ and colony-forming units previously standard counted using the AGAR plate method. Different concentrations of *L. plantarum* were inoculated to achieve the desired CFU levels in the coating solution [15].

### 2.3. Preparation of Water-in-Water Pickering Emulsion

Following the method described by Zhang et al. [16], the water-in-water Pickering emulsion was prepared with slight modifications. A stock solution was prepared by dissolving 8% HPMC and 20% MD in distilled water. Subsequently, *Lactobacillus plantarum* Y-35 suspensions at concentrations of 1 × 10^10^, 1 × 10^8^, and 1 × 10^6^ CFU/g were dispersed in HPMC solution. MD solution was then added, and the mixture was magnetically stirred at 800 rpm for 20 min to form W/W Pickering emulsions with varying bacterial concentrations. In these emulsions, maltodextrin (MD) served as the dispersed phase while hydroxypropyl methylcellulose (HPMC) constituted the continuous phase. These emulsions were named W/W-PE, W/W-PL^10, W/W-PL^8, and W/W-PL^6. W/W-PE was a water-in-water Pickering emulsion, W/W-PL^10 was an emulsion with 1 × 10^10^ (CFU) g^−1^
*Lactobacillus* suspension added, W/W-PL^8 was an emulsion with 1 × 10^8^ (CFU) g^−1^
*Lactobacillus* suspension added, and W/W-PL^6 was an emulsion with 1 × 10^6^ (CFU) g^−1^
*Lactobacillus* suspension added.

### 2.4. Survival Rate of Lactobacillus Y-35

Following the method described by Sánchez-González et al. [17], the determination was carried out with slight modifications. The W/W Pickering emulsion containing Lactobacillus was serially diluted in tryptone phosphate water and inoculated onto MRS agar. The bacterial activity was then quantified using the plate counting method. Measurements were performed in triplicate every two days until day 8. The results were expressed as the logarithm of colony-forming units per milliliter of solution (log CFU/mL).

### 2.5. W/W Pickering Emulsion Characterization

#### 2.5.1. Storage Stability of Pickering Emulsion

The microstructure of the emulsion droplets was observed using an optical microscope with a 400 objective lens (Discovery.V20, ZEISS, Oberkochen, Germany). A coverslip was placed over the emulsion after it was deposited onto the glass slide [18]. Emulsion storage stability was assessed through visual observation. All observations were conducted at 25 °C.

#### 2.5.2. The Particle Size and Zeta Potential of Solid Particles and Pickering Emulsion Particles

Dynamic Light Scattering (DLS) (Nano ZSE, Malvern Instruments, Malvern, UK) was utilized to determine the particle size distribution and zeta potential of the suspensions at 25 °C. The Pickering emulsion was diluted with deionized water at a ratio of 1:100 (*v*/*v*) relative to the solid particles to minimize multiple scattering effects [19].

### 2.6. Biocomposite Films Characterization

The prepared Pickering emulsion was uniformly cast onto a polyethylene plate (10 cm × 10 cm) and dried in a forced-air oven (CS101-3, Chongqing, China) at 45 °C for 12 h to form films [20]. Based on the *L. plantarum* concentration, the films were labeled as W/W-PE, W/W-PL^10, W/W-PL^8, and W/W-PL^6. These emulsion-derived films were used for all subsequent characterization.

#### 2.6.1. Scanning Electron Microscopy (SEM)

We utilized scanning electron microscopy to characterize the emulsion-based films’ microstructure (JSM-7500F, JEOL, Tokyo, Japan). The samples were sputter-coated with a conductive layer (e.g., gold or platinum) under vacuum conditions to enhance surface conductivity, and imaging was performed at an acceleration voltage of 15 kV.

#### 2.6.2. Fourier Transform Infrared Spectroscopy (FTIR)

Fourier Transform Infrared (FTIR) analysis was conducted using a Thermo Fisher Scientific (Nicolet iS5, Waltham, MA, USA) spectrometer in the wavenumber range of 400–4000 cm^−1^, with accumulation of 32 scans (4 cm^−1^ resolution) [21].

#### 2.6.3. X-Ray Diffraction (XRD)

The crystalline structure of the thin films was characterized by X-ray diffraction (Ultima IV-185, Rigaku Co., Ltd., Tokyo, Japan) using Cu Kα radiation (λ = 1.5406 Å). Tests were carried out at 25 °C with a 2θ scanning range of 2–60° at a scan rate of 2°/min. Each sample was analyzed in triplicate to ensure reproducibility [21].

#### 2.6.4. Differential Scanning Calorimetry (DSC)

The thermal behavior of the films was analyzed using a differential scanning calorimeter (DSC Q200F3, TA Instruments, New Castle, DE, USA). Samples (~2 mg) were encapsulated in hermetically sealed aluminum crucibles and subjected to heating and cooling cycles at a constant rate of 10 °C/min under a nitrogen purge (flow rate: 50 mL/min). The temperature program consisted of dynamic scans from 25 °C to 500 °C to evaluate phase transitions and thermal stability.

#### 2.6.5. Thermogravimetric Analysis (TGA)

Film samples were subjected to thermogravimetric analysis using a TA Instruments Q500 (New Castle, DE, USA) at a heating rate of 10 °C/min under a nitrogen atmosphere. Samples (~10 mg) were loaded into aluminum crucibles, and the temperature was ramped from 25 °C to 600 °C [22].

#### 2.6.6. Water Vapor Permeability (WVP)

The WVP of the antibacterial membrane was determined using a water vapor transmission rate tester (Systester Instruments Co., Ltd., Jinan, China). Testing occurred at standard conditions of 25 °C temperature and 90% humidity, following a continuous measurement period of 25 h.

#### 2.6.7. Antioxidant Properties of the Membrane

The coating sample (1 g) was weighed and mixed with 10 mL of anhydrous ethanol. The mixture was vortexed for 30 min and then centrifuged at 4000 rpm and 4 °C for 10 min. The supernatant was collected for subsequent analysis.

We determined DPPH radical scavenging rates employing Ren et al.’s procedure [23], with slight modifications. Briefly, the supernatant was diluted 10-fold with anhydrous ethanol. Next, 2 mL of the diluted sample was mixed with 2 mL of 0.2 mM DPPH solution, incubated in the dark for 30 min, and then measured at 517 nm using a UV-Vis spectrophotometer.

The ABTS radical scavenging assay was conducted following the method of Zhang et al. [24]. A 50 μL supernatant aliquot was added to 3 mL ABTS solution, incubated for 6 min, and measured at 714 nm. The radical scavenging activity was calculated using the following Equation (1):(1)DPPH/ABTS radical scavenging activity (%) = [(A _control_ − Aᵢ)/A _control_] × 100

*A_control_* refers to the absorbance value of the blank group, and *A_i_* refers to the absorbance value of the experimental group.

#### 2.6.8. Antibacterial Activity of the Material

The antibacterial activity of the membrane solution against Gram-positive bacteria (*Staphylococcus aureus*) and Gram-negative bacteria (*Escherichia coli*) was evaluated using the Oxford cup method. Bacterial suspensions were spread onto LB agar plates for culture. After the agar-containing bacteria had solidified, 200 μL of the film extract was added to the Petri dish through an Oxford cup (stainless steel cylinder with an outer diameter of 8.0 mm, an inner diameter of 6.0 mm, and a height of 10.0 mm). The plates were then incubated at 37 °C for 24 h. The diameter of the inhibition zone around the film sample was measured in triplicate using a caliper [25].

The antibacterial activity of the membrane solution against *Alternaria alternata* was investigated using the agar diffusion method. After preparing the PDA culture medium, multiple concentrations of the membrane solution were mixed with the PDA for later utilization. Following a 10-day culture period, mycelial plugs of *Alternaria* were obtained using a 12 mm diameter puncher. These plugs were then inoculated onto PDA plates containing different concentrations of the membrane solution. The plates were subjected to 7 days of growth at 26 ± 1 °C under controlled incubation conditions. The colony diameters were measured based on the mycelial growth conditions.

#### 2.6.9. Cell Membrane Integrity

Transfer 100 μL of the washed spore suspension of *Alternaria alternata* into 100 mL potato dextrose broth with subsequent shaking incubation (130 rpm) at 28 °C for 72 h. After incubation, collect the mycelia and then extensively with sterile water for complete medium removal. Resuspend 0.2 g of the mycelia in different concentrations of emulsions and incubate at 130 rpm and 28 °C for 12 h. Centrifuge the mycelial culture at 4 °C (5000× *g*, 10 min), discard the supernatant, and rinse the mycelia with PBS three times. After thorough rinsing, samples were centrifuged and finally stained with PI for visualization [26].

### 2.7. Postharvest Indicators of Cherry Tomatoes

#### 2.7.1. Cherry Tomato Coating Treatment and Visual Quality Assessment

One hundred mature cherry tomatoes of uniform size were selected, from which fresh, undamaged, and disease-free fruits were randomly divided into five treatment groups. The selected fruits were washed with distilled water and air-dried. The control group was soaked in distilled water, while the remaining four groups were treated with W/W-PE, W/W-PL^10, W/W-PL^8, and W/W-PL^6 coating solutions, respectively. Each group was immersed in the corresponding Pickering emulsion coating solution for 1 min and then allowed to air-dry. Cherry tomatoes were aseptically wounded with a 2 mm diameter sterile needle. Fruits were inoculated with 10 μL fungal suspension (*A. alternata*, 10^6^ CFU/mL) per wound and dried at 25 ± 2 °C. The fruits were stored at 25 ± 1 °C and approximately 80% RH for 8 days. Sampling was conducted at five time points (0, 2, 4, 6, and 8 days) over the 8-day period.

Furthermore, the spoilage rate of cherry tomatoes was evaluated daily by a panel of 10 trained researchers. Decay severity was graded on a 0–4 scale: 0 (no decay), 1 (1–25% affected), 2 (26–50%), 3 (51–75%), and 4 (76–100%). Visual quality was simultaneously assessed using a 9-point hedonic scale (1 = extremely poor to 9 = excellent) considering color, gloss, and structural integrity.

#### 2.7.2. Decay Rate

The decay rate of cherry tomatoes was monitored and documented through photographic records based on the extent of decay observed post-harvest [27]. The decay rate was calculated using the following Formula (2):(2)Decay rate (%) = (a/b) × 100 where *a* represents the number of decayed fruits, and *b* represents the total number of fruits. The decay rate was assessed every two days throughout the experiment.

#### 2.7.3. Hardness

A texture analyzer (TA. XT. plus, Stable Micro Systems, Godalming, UK) was used to measure the hardness of the samples. A P/2 probe was employed to puncture the sample at the equatorial center region of cherry tomatoes at a test speed of 1 mm/s, with a strain threshold set at 40% [28].

#### 2.7.4. Weight Loss Rate

The weight loss rate was calculated as the percentage reduction in fruit weight after storage relative to the initial weight before storage [29]. The weight of each group of treated samples was measured using a 0.1 mg precision analytical balance and recorded separately. The initial weight of the samples was recorded as w_1_, and the weights on days 2, 4, 6, and 8 were recorded as w_2_. The weight of the empty container was recorded as w_0_. Calculate the rate of weightlessness according to the following Formula (3):(3)Weight loss rate (%) = [(w_1_ − w_2_)/(w_1_ − w_0_)] × 100

#### 2.7.5. Electrolyte Leakage

To assess membrane integrity alterations, electrolytic conductivity was measured as a permeability indicator. Following a modified method based on Meiyan Zhu et al. [30], 50 mL of distilled water was added to a beaker, and 25 g of cherry tomatoes were placed into the beaker. The beaker was kept at room temperature for 20 min, and the initial conductivity (E_0_) of the solution was measured. Subsequently, the beaker containing the sample was boiled for 15 min and then cooled to 25 °C, after which the final conductivity (E_1_) was measured. Each treatment group was tested in triplicate. The electrolyte leakage (%) was calculated using the following Formula (4):(4)Electrolyte leakage (%) = (E_0_/E_1_) × 100

#### 2.7.6. Soluble Solids Content

The soluble solids content (SSC) in the stored samples was measured with a digital refractometer (WYT-1, Qingyang Optics Instrument Co., Ltd., Chengdu, China). For each sample, approximately three drops of juice were deposited on the refractometer prism to measure soluble solids concentration (%).

#### 2.7.7. Titratable Acidity

A 10 g sample was weighed and homogenized. The homogenized sample was then transferred to a 100 mL volumetric flask, and the volume was made up to the mark within 30 min. The solution was filtered through gauze, and 20 mL of the filtrate was collected and transferred to a conical flask. To the filtrate, 2 drops of phenolphthalein were added, then 0.1 M NaOH was gradually added until reaching a persistent pink hue. [31]. Titratable acidity (TA) was determined according to the Formula (5):(5)Titratable Acidity (%) = (V × c × (V_1_ − V_0_) × f)/(Vₛ × m) × 100 where V is the sample extract’s total volume (mL), Vₛ the filtrate volume for titration (mL), c is the NaOH titrant’s concentration (mol/L), V_1_ the NaOH volume used in filtrate titration (mL), V_0_ the NaOH volume for blank titration (mL), m the sample mass (g), and f the conversion factor (g/mmol).

#### 2.7.8. Ascorbic Acid Content

The vitamin C content was determined according to the national standard (GB 5009.86-2016) using the 2,6-dichlorophenol titration method [32]. A 10 g sample was weighed and homogenized with 10 mL of 2% oxalic acid solution. The homogenate was then diluted to 50 mL with distilled water. A 5 mL aliquot of the extract was transferred to a test tube, and the solution was titrated with 2,6-dichlorophenol reagent until the color changed from red to blue, persisting for at least 15 s. The volume of the reagent consumed was recorded. Ascorbic acid concentration was determined according to Equation (6):(6)Ascorbic Acid Content (mg/100 g) = [(V × (V_1_ − V_0_) × ρ)/(m × Vₛ)] × 100 where V is the total sample extract volume (mL), V_1_ and V_0_ represent the volumes of 2,6-dichlorophenol reagent consumed during sample and blank titrations (mL), respectively, ρ denotes the ascorbic acid equivalent concentration (mg/mL), Vₛ is the aliquot volume used for titration (mL), and m is the sample mass (g). All analyses were performed in triplicate.

#### 2.7.9. Malondialdehyde Content

MDA concentration was quantified through the TBA reaction method, following a modified protocol based on Shang et al. [33]. Briefly stated, 1 g of cherry tomato powder was homogenized with 10 mL of 10% (*w*/*v*) trichloroacetic acid (TCA) and centrifuged at 4000× *g* for 10 min at 4 °C. The supernatant was collected as the MDA extract. A 2 mL aliquot of the supernatant was mixed with 2 mL of 0.6% (*w*/*v*) TBA solution, heated in a boiling water bath for 15 min, and then rapidly cooled. The absorbance of the mixture was measured at 450 nm, 532 nm, and 600 nm using a spectrophotometer. MDA concentration was determined according to Equation (7):(7)MDA Content (μmol/g FW) = 6.45 × (A_532_ − A_600_) − 0.56 × A_450_ where A532 is the absorbance at 532 nm, A600 is the absorbance at 600 nm, and A450 is the absorbance at 450 nm.

### 2.8. Statistical Analysis

Data were organized and visualized using Origin 2021 (Origin Lab Corporation, Northampton, MA, USA). Statistical analysis was performed using SPSS Statistics 26.0 (IBM Corp., Armonk, NY, USA). One-way analysis of variance (ANOVA) followed by Duncan’s multiple range test was conducted to assess the significance of differences between means (*p* < 0.05).

## 3. Results and Discussion

### 3.1. Activity of Probiotics in Emulsion

As shown in Figure 1, the control group (pure *L. plantarum* in MRS medium) exhibited no significant changes during storage, as the MRS medium provided adequate nutrients, ensuring stable viability. In contrast, the survival rate of *L. plantarum* encapsulated in the Pickering emulsion declined gradually over the 8-day storage period with varying additive concentrations, though the bacterial concentration remained consistently high. Storage temperature and environmental conditions are critical factors influencing cell survival rates [34]. This observation may be attributed to the additional nutrients provided by hydroxypropyl methylcellulose (HPMC) and maltodextrin (MD), which supported the strain’s viability. This finding is consistent with previous studies indicating that the survival rate of this strain is not significantly affected by substrate composition or storage conditions, as reported by Sánchez-González et al. [17].

### 3.2. Characterization of Pickering Emulsion

#### 3.2.1. Microscopic Structure of the Emulsion

As shown in Figure 2A, W/W-PE droplets lacking solid particles exhibit the largest diameter and lowest stability, compromising emulsion stabilization and potentially causing phase separation. In contrast, the droplets in the Pickering emulsion containing probiotics showed variations in diameter depending on the concentration of *L. plantarum*. Among the tested formulations, the emulsion with the W/W-PL^8 concentration demonstrated the highest stability, with droplets exhibiting a more uniform dispersion. This optimal stability likely results from balanced bacterial adsorption at the interface, as evidenced by the zeta potential (−27.6 mV) values (Table 1), which indicate effective electrostatic and steric stabilization. Figure 2B displays the Pickering emulsion after 15-day storage, showing homogeneous dispersion without sedimentation, demonstrating excellent stability across all emulsion groups.

#### 3.2.2. Particle Size and Zeta Potential of Solid Particles and Pickering Emulsions

As shown in Figure 3, the particle size of *L. plantarum* solid particles was 2379.3 nm, with a polydispersity index (PDI) of 0.129. A lower PDI value indicates a more stable and uniform emulsion [35]. Therefore, the addition of *L. plantarum* to Pickering emulsions enhances their stability. Stable Pickering emulsions in this work were characterized by zeta potentials with absolute values above 30 mV. [36]. Table 1 shows that in the Pickering emulsion, when the concentration of *L. plantarum* was 1 × 10^8^ CFU/g, the zeta potential reached −27.6 mV. Table 1 shows that in the Pickering emulsion with 1 × 10^8^ CFU/g *L. plantarum*, the zeta potential reached −27.6 mV, approaching the stability threshold. This near-−30 mV value suggests electrostatic stabilization dominates at optimal bacterial concentration, while steric hindrance from bacterial surface proteins contributes additional stability at lower concentrations, demonstrating higher stability compared to other experimental groups. This phenomenon might be caused by the fact that the other experimental groups deviated from the optimal bacterial concentration (either too high or too low), thereby disrupting the interface balance of the emulsion.

### 3.3. Characterization of Coating Solution

#### 3.3.1. Scanning Electron Microscopy (SEM) Analysis

Figure 4 shows the planar and cross-sectional morphologies of the films. The W/W-PE group exhibited a relatively smooth surface morphology compared to the other groups, with no visible protrusions or pores, indicating better structural integrity. In contrast, the films containing *L. plantarum* displayed irregular, non-continuous “island-like” structures with a roughly circular shape compared to the control group. The structural characteristics of the films varied depending on the concentration of the bacterial suspension added. The W/W-PL^10 and W/W-PL^8 groups showed more surface protrusions than the control group, suggesting that *L. plantarum* was successfully embedded into the film matrix during the drying process, leading to increased surface roughness. These morphological changes indicate that bacterial incorporation altered the polymer network organization, likely through interactions between bacterial surface proteins and polysaccharide chains. The W/W-PL^6 group, which contained fewer bacteria, exhibited a surface flatness similar to that of the control group. Cross-sectional analysis revealed that the films with added bacteria were significantly thinner than the control group, likely due to differences in density. The control group exhibited a distinct multi-layered structure with varying color intensities, whereas the films with added bacteria showed altered aggregation patterns of the film-forming substances, possibly due to their improved solubility in the film matrix. These observations further confirm that the W/W-PL^8 group demonstrated superior stability, resulting in a more uniform and compact film structure that provides an effective physical barrier.

#### 3.3.2. Fourier Transform Infrared Spectroscopy (FTIR) Analysis

Figure 5 presents the FTIR spectral profiles of the fabricated films. Both hydroxypropyl methylcellulose (HPMC) and maltodextrin (MD) exhibited characteristic peaks at 3500 cm^−1^, 2900 cm^−1^, 1653 cm^−1^, and 1049 cm^−1^. The broad absorption peak at 3350 cm^−1^ is attributed to the O-H stretching vibration, while the peak near 2900 cm^−1^ corresponds to the C-H stretching vibration [37]. The peak at 1653 cm^−1^ is associated with the C=O stretching vibration of amide I [38]. The strong absorption peak at 1049 cm^−1^ is likely due to the asymmetric stretching vibration of the ether bond (C-O-C) in the cellulose structure. Additionally, the peak at 991 cm^−1^ in maltodextrin is typically assigned to the C-O-C stretching vibration in carbohydrates. After the addition of different concentrations of *L. plantarum*, no new peaks were observed. However, the originally broad peaks became sharper, which may be attributed to the metabolic activity of *L. plantarum*. It is possible that impurities in the sample initially broadened the characteristic peaks, and *L. plantarum* metabolized these impurities during incubation, resulting in a “purer” sample with reduced interference. This led to sharper characteristic peaks. These findings suggest that the addition of *L. plantarum* did not form covalent bonds with HPMC or MD, nor did it induce any significant interactions between them. This observation is consistent with the results reported by Joana et al. [39].

#### 3.3.3. X-Ray Diffraction (XRD)

As shown in Figure 6, the XRD patterns reveal that the HPMC film exhibits two relatively broad diffraction peaks at approximately 7° and 20° (2θ), corresponding to its characteristic semi-crystalline structure, paralleling the findings of Zhang et al. [40]. Similarly, MD displays a distinct crystalline peak at 20° (2θ), indicative of its short-range ordered molecular packing. Notably, the absence of sharp diffraction peaks in MD suggests its low crystallinity degree (~15%), which facilitates homogeneous dispersion within the HPMC matrix. The W/W-PE film shows a diffraction profile analogous to HPMC, with no additional peaks detected. Upon incorporation of probiotics at varying concentrations into W/W-PE, neither peak position nor intensity exhibited significant alterations, confirming that the probiotic addition did not induce structural modifications.

#### 3.3.4. Differential Scanning Calorimetry (DSC)

To investigate the thermal properties of the composite films, differential scanning calorimetry (DSC) was performed. As shown in Figure 7, the endothermic and exothermic transitions were analyzed, where higher peak temperatures correspond to enhanced thermal stability [41]. As evidenced by the DSC thermograms, all experimental film samples exhibited a distinct exothermic peak at 225 °C, corresponding to heat release during thermal decomposition. Concurrently, an endothermic peak was observed at 303 °C, indicative of heat absorption during phase transition. The peak enthalpy values remained statistically invariant across all probiotic concentrations, confirming that *L. plantarum* incorporation neither disrupted the polymeric network nor introduced thermally labile components. This thermal robustness aligns with the FTIR observations of preserved molecular interactions. These results demonstrate that probiotic incorporation maintains the thermal stability of the composite films, as no significant alterations in thermal transitions were detected.

#### 3.3.5. Thermogravimetric Analysis (TGA)

Thermogravimetric analysis (TGA) is an important technique for evaluating the thermal stability, composition, and degradation kinetics of polymer materials, blends, and composites [42]. As shown in Figure 8, the TGA curves show high overlap among all tested film samples. In the initial stage, only minor mass loss occurs, likely due to dehydration and evaporation of volatile components. Significant thermal degradation is observed between 200 °C and 400 °C, where the curves show rapid decline, corresponding to the decomposition of organic components and polymer chains. The W/W-PE composite exhibits different thermal degradation characteristics compared to pure HPMC and MD, indicating modified thermal behavior after blending. This change may result from intermolecular hydrogen bonding interactions affecting thermal stability. Furthermore, different probiotic concentrations influence thermal stability—higher concentrations lead to reduced stability, while lower concentrations show less pronounced effects.

**Figure 8 foods-14-02729-f008:**
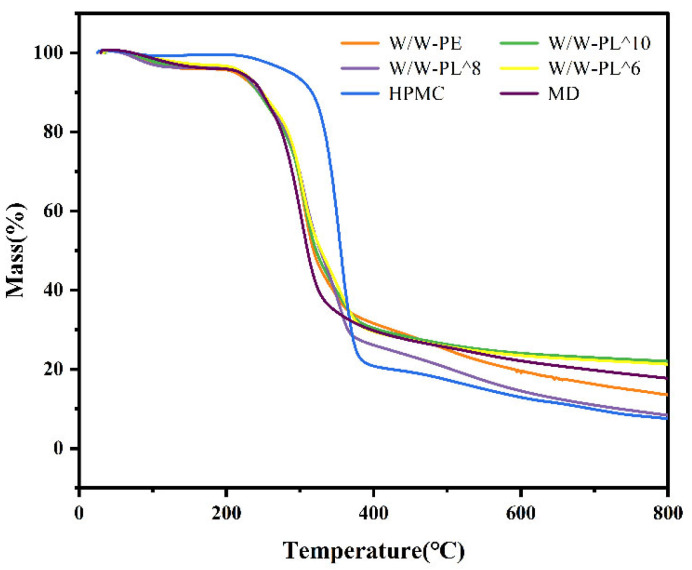
The TGA thermogram of the film.

#### 3.3.6. Water Vapor Transmission Rate (WVP)

As shown in Figure 9, under conditions of 25 °C and 90% RH, the WVP of the film materials exhibited an exponential increase. The W/W-PL^10 film demonstrated the lowest WVP, second only to the W/W-PE group. This phenomenon may be attributed to the concentration of *L. plantarum*, which enhanced the barrier properties of the film. As the concentration of *L. plantarum* decreased, the WVP of the films gradually increased. This trend may be due to the looser film structure resulting from the addition of bacteria.

**Figure 9 foods-14-02729-f009:**
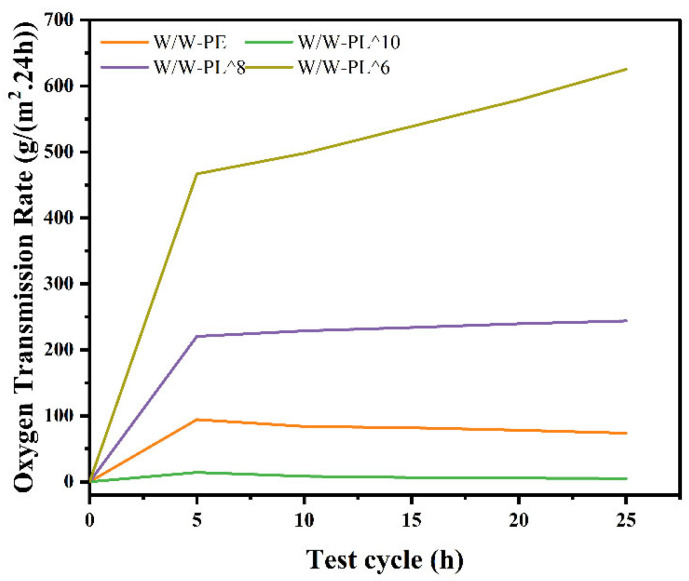
Water vapor permeability of thin films.

#### 3.3.7. Antioxidant Properties of the Material

The antioxidant properties of food packaging films are crucial for food preservation, as they can mitigate oxidative damage and delay aging. As shown in Figure 10, the emulsion without added bacteria exhibited the weakest scavenging activity, likely due to the lack of significant free-radical scavenging ability in hydroxypropyl methylcellulose (HPMC) and maltodextrin (MD). However, the antioxidant capacity varied with the concentration of *L. plantarum* in the emulsion. Li et al. also reported that the addition of probiotics to composite films significantly enhanced antioxidant activity [43]. The W/W-PL^8 group demonstrated the strongest scavenging ability, likely because it reached the optimal bacterial concentration within the tested range. These findings identify an optimal probiotic concentration (10^8^ CFU/mL) for active packaging, balancing viability and function. Exceeding this range may lead to an inhibitory effect, resulting in reduced antioxidant activity. This phenomenon is related to the growth status of the bacterial strain, the accumulation of metabolites, and potential competitive interactions. The DPPH free-radical scavenging abilities of the W/W-PL^10 and W/W-PL^6 groups were 89% and 66%, respectively, which can be attributed to excessive and insufficient bacterial concentrations. Additionally, the ABTS free-radical scavenging activity increased with increasing *Lactobacillus plantarum* concentration, reaching 49%, 58%, and 63%, respectively.

**Figure 10 foods-14-02729-f010:**
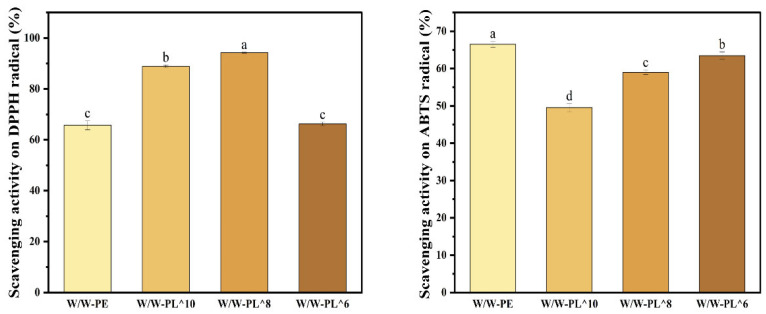
Radical scavenging capacity of thin films assessed by DPPH and ABTS assays. Means in each cepa followed by different lowercase letters indicate significant differences (*p* < 0.05) according to Duncan’s test.

#### 3.3.8. Antibacterial Activity of the Materials

In this study, the Oxford cup diffusion method was utilized to determine the antimicrobial activity of Pickering emulsions toward both Gram-negative (*E. coli*) and Gram-positive (*S. aureus*) bacteria. The emulsions containing *L. plantarum* exhibited significant antibacterial activity. As shown in Figure 11A,B, the size of the inhibition zones varied depending on the concentration of *L. plantarum* in the emulsions. The W/W-PL^10 group demonstrated the strongest antibacterial effect, followed by W/W-PL^8 and W/W-PL^6. These results indicate that higher concentrations of the bacterial suspension enhance the inhibitory effect on both Gram-positive and Gram-negative bacteria. Furthermore, the inhibition zone diameter for *Staphylococcus aureus* was significantly larger than that for *Escherichia coli*. This difference can be attributed to the bacteriocins produced by *L. plantarum*, which disrupt the cell membrane of *Staphylococcus aureus*, leading to leakage of cellular contents and subsequent bacterial death.

In Figure 12A, it can be observed that the emulsion exhibits an inhibitory effect on *Alternaria alternata*. Notably, the emulsion containing *Lactobacillus* demonstrated enhanced antibacterial efficacy, which correlated with the bacterial concentration. In the control group, *Alternaria alternata* colonies expanded to cover the entire PDA plate, reaching a colony diameter of 10.9 cm. After 7 days of incubation, the inhibition zone diameter of the W/W-PE group (without *Lactobacillus*) was 7.5 cm, while the W/W-PL^10, W/W-PL^8, and W/W-PL^6 groups (with *Lactobacillus*) exhibited inhibition zone diameters of 3 cm, 3.3 cm, and 5.5 cm, respectively. Compared to the control group, the *Lactobacillus*-containing emulsions exhibited antibacterial efficacy rates of 72.48%, 69.72%, and 49.54%.

**Figure 11 foods-14-02729-f011:**
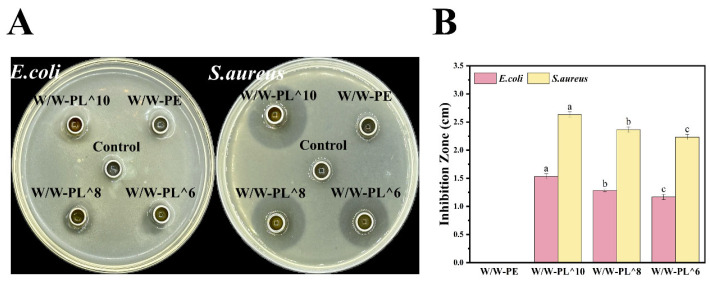
Antibacterial efficacy of the emulsion against *E. coli* and *S. aureus* within 12 h (**A**), and the diameter of the inhibition zone (**B**). Means in each cepa followed by different lowercase letters indicate significant differences (*p* < 0.05) according to Duncan’s test.

**Figure 12 foods-14-02729-f012:**
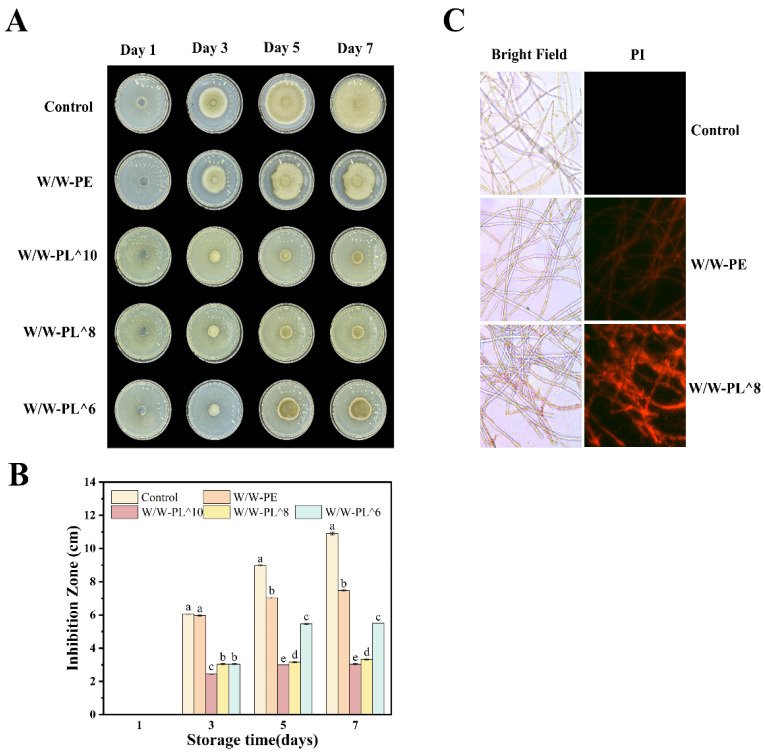
Antibacterial effect of the emulsion on *Alternaria alternata* (7 days) (**A**), diameter of the inhibition zone (**B**), and integrity of the cell membrane (**C**). Means in each cepa followed by different lowercase letters indicate significant differences (*p* < 0.05) according to Duncan’s test.

#### 3.3.9. Cell Membrane Integrity

Propidium iodide (PI) staining is a widely used method to assess cell membrane integrity. The red fluorescent dye PI is membrane-impermeable and can only penetrate cells with compromised membranes, subsequently binding to nucleic acids [44]. By analyzing the fluorescence intensity of *Alternaria alternata* mycelia, the impact of *L. plantarum* emulsions (W/W-PE and W/W-PL^8 concentrations) on the cell membrane permeability of *Alternaria alternata* was evaluated. As shown in Figure 12C, untreated control cells exhibited negligible fluorescence. In contrast, emulsion-treated groups displayed increased fluorescence intensity, with significantly enhanced red fluorescence observed in bacterial cells. Notably, the W/W-PL^8 treatment group demonstrated the strongest fluorescence signal, indicating severe membrane damage and loss of integrity caused by *Lactobacillus plantarum*. This membrane disruption effectively inhibited bacterial growth.

### 3.4. Effects of Coatings on the Physiology and Biochemistry of Cherry Tomatoes

#### 3.4.1. Visual Evaluation of Cherry Tomato Quality and Decay Rate

Sensory evaluation is one of the key criteria for assessing fruit quality. After the treated cherry tomatoes were inoculated with *Alternaria alternata* and stored at 25 °C for 8 days, a panel of 10 trained researchers conducted daily visual assessments. Storage photographs (Figure 13) and quantitative evaluations demonstrated that the W/W-PL^8 group showed optimal preservation with a visual score of 8.1 ± 0.3 and decay grade of 0.4 ± 0.1, exhibiting intact pore structure and no obvious attachment of microorganisms. Moderate protection was observed in the W/W-PE group (visual score: 6.3 ± 0.3; decay: 1.6 ± 0.2), while the W/W-PL^10 treatment showed limited efficacy (visual score: 5.7 ± 0.5; decay: 2.3 ± 0.3) with partial fungal colonization. The W/W-PL^6 group demonstrated significant quality deterioration (visual score: 4.9 ± 0.4; decay: 2.8 ± 0.2), and the uncoated control samples developed severe decay (visual score: 3.2 ± 0.6; decay: 3.5 ± 0.4) with widespread lesions.

The Pickering coating solution containing different concentrations of *Lactobacillus* was tested for its antibacterial effect in vitro. The results showed that the higher the bacterial content, the stronger the antibacterial effect. However, on fruits, higher bacterial content might lead to greater damage. Conversely, lower bacterial content resulted in a weaker protective effect on the fruits. Therefore, selecting an optimal bacterial concentration is crucial. In this study, the W/W-PL^8 concentration was identified as the most suitable, offering the best preservation effect for cherry tomatoes. This optimal concentration not only effectively inhibited pathogen growth but also maintained fruit integrity, suggesting a delicate balance between probiotic activity and host tissue compatibility. The findings highlight the importance of strain- and dose-specific optimization in probiotic-based preservation technologies.

#### 3.4.2. Hardness and Weight Loss

During storage, the postharvest quality of cherry tomatoes is vulnerable to diverse external influences, particularly microbial infection and nutrient loss, leading to weight reduction and fruit softening. As illustrated in Figure 14B, cherry tomatoes showed diminishing firmness over time. This phenomenon may be attributed to structural changes in the internal cell matrix of the tomato pulp, including pectin degradation and cell membrane rupture [32]. The W/W-PL^8 coating reduced the hardness loss rate by 54% compared to the control (*p* < 0.05), likely due to bacterial metabolites inhibiting cell wall-degrading enzymes. Among the groups, the W/W-PL^8 group showed a slower decline in hardness, indicating that *L. plantarum* effectively maintained the texture of the tomatoes.

As shown in Figure 14C, significant differences in weight loss were observed between the control group and the experimental groups by the fourth day of storage (*p* < 0.05). By the sixth day, the control group and the W/W-PE group reached weight loss rates of 5.93% and 5.62%, respectively. On the eighth day, the control group exhibited a weight loss of 8.06%, while the W/W-PE and W/W-PL^8 groups showed weight losses of 7.26% and 6.19%, respectively. The W/W-PL^8 group demonstrated a 9.93% and 23.2% reduction in weight loss compared to the control group. These results indicate that the control group experienced the most severe weight loss, whereas the W/W-PL^8 group effectively minimized weight reduction.

#### 3.4.3. Soluble Solids, Titratable Acidity, and Vitamin C Content

The contents of soluble solids, TA, and vitamin C (VC) are critical indicators for evaluating post-harvest storage quality and fruit flavor, as well as key determinants of consumer preference [45]. As shown in Figure 14D, the SSC exhibited an overall declining trend during storage. The initial SSC of the fruit was 8.6%, and by the end of the storage period, the W/W-PL^8 group maintained a higher SSC (6.9%) compared to the control group (5.0%) and the W/W-PE group (6.4%). This improved SSC retention in the W/W-PL^8 group (1.9% higher than control) aligns with its better performance in reducing weight loss (Figure 14C), suggesting the coating’s physical barrier effect contributes to sugar preservation. This decline may be attributed to the continuous respiration of the fruit, which consumes a significant amount of sugar, leading to a gradual reduction in SSC.

TA content, which reflects the sugar–acid ratio and influences fruit taste, is significantly affected by metabolic rates, particularly respiration [46]. As shown in Figure 14E, the TA content of tomatoes during storage initially increased and then decreased. This trend may be due to an early respiration peak followed by a decline. During respiration, organic acids are metabolized, resulting in a gradual decrease in TA content [47]. The decline in TA content was slower in the treatment groups compared to the control group.

As shown in Figure 14F, the VC content of tomatoes decreased with prolonged storage. This reduction is likely caused by the reaction of VC oxidase in cherry tomatoes with atmospheric oxygen, leading to the degradation of VC into dehydroascorbic acid, which loses its physiological activity [48]. The VC content in the control group decreased from 63 mg/100 g initially to 35 mg/100 g, while the best-performing treatment group (W/W-PL^8) only decreased to 44 mg/100 g. This improved VC retention (25.7% higher than control at endpoint) is consistent with the coating’s demonstrated ability to reduce moisture loss (Figure 14C), suggesting the barrier properties may help maintain cellular integrity and nutrient retention.

#### 3.4.4. Electrolyte Leakage and Malondialdehyde (MDA) Content

Relative conductivity is an indicator of membrane integrity, with lower values typically indicating better membrane stability. As shown in Figure 14G, all groups exhibited an increasing trend in relative conductivity during storage. However, the W/W-PL^8 treatment group showed significantly lower conductivity compared to the other treatment groups and the control group (*p* < 0.05). After the second day, the conductivity of the W/W-PL^8 group stabilized at 38%, while the control group (CK) and W/W-PE group reached 52% and 43%, respectively. The 26.9% lower conductivity in W/W-PL^8 compared to the control correlates with its reduced MDA content, demonstrating coordinated protection of membrane integrity.

Malondialdehyde (MDA) is a key product of membrane lipid peroxidation in plant cells and is widely used as an indicator of cell membrane damage [49]. Figure 14H demonstrates a progressive accumulation of MDA in all experimental groups throughout storage. Both the W/W-PL^8 and W/W-PE treatment groups exhibited lower MDA content than the control group, demonstrating that the film coating containing Pickering emulsion and *L. plantarum* effectively slowed membrane lipid peroxidation and reduced MDA accumulation in fruit cells.

## 4. Conclusions

In this study, the concentration of *L. plantarum* Y-35 was optimized as an emulsion stabilizer, and the resulting W/W-PE emulsion was used to coat cherry tomatoes for preservation. The results demonstrated that the probiotic emulsion with a W/W-PL^8 concentration exhibited the best stabilization effect, achieving a zeta potential of −27.6 mV and maintaining high probiotic activity throughout the storage period. Additionally, the material demonstrated a free-radical scavenging capacity of up to 94% and exhibited significant antibacterial activity against *Escherichia coli* and *Staphylococcus aureus*, with inhibition zone diameters of 1.28 cm and 2.37 cm, respectively. Moreover, the W/W-PL^8 treatment showed remarkable preservation effects on cherry tomatoes. It not only effectively slowed the decay rate during storage but also maintained fruit firmness (95.5 N) and controlled the weight loss rate (6.19%). Furthermore, this treatment significantly reduced the respiratory rate (0.009 mg/kg/h), electrolyte leakage (39.76%), and malondialdehyde (MDA) accumulation, thereby delaying the fruit’s senescence. Throughout storage, the treatment group showed significantly reduced fluctuations in SSC, TA, and VC levels relative to controls, demonstrating superior cherry tomato quality preservation. Although this study has confirmed that the stable water-in-water Pickering emulsion of *Lactobacillus plantarum* has an excellent preservation effect on cherry tomatoes, the current system still has the problem that the stability of the emulsion is greatly affected by environmental factors (such as temperature and pH). Future research can focus on addressing this issue. Additionally, this emulsion coating technology can be extended to the preservation of more fruits and vegetables, in order to reduce post-harvest losses of fresh agricultural products and solve the scale challenges in commercial production.

## Figures and Tables

**Figure 1 foods-14-02729-f001:**
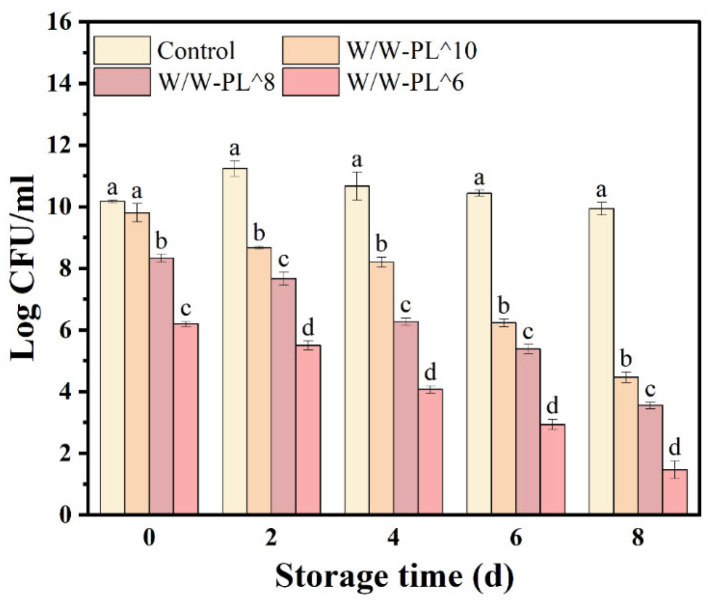
The survival rate of *Lactobacillus plantarum* in the Pickering emulsion. Means in each cepa followed by different lowercase letters indicate significant differences (*p* < 0.05) according to Duncan’s test.

**Figure 2 foods-14-02729-f002:**
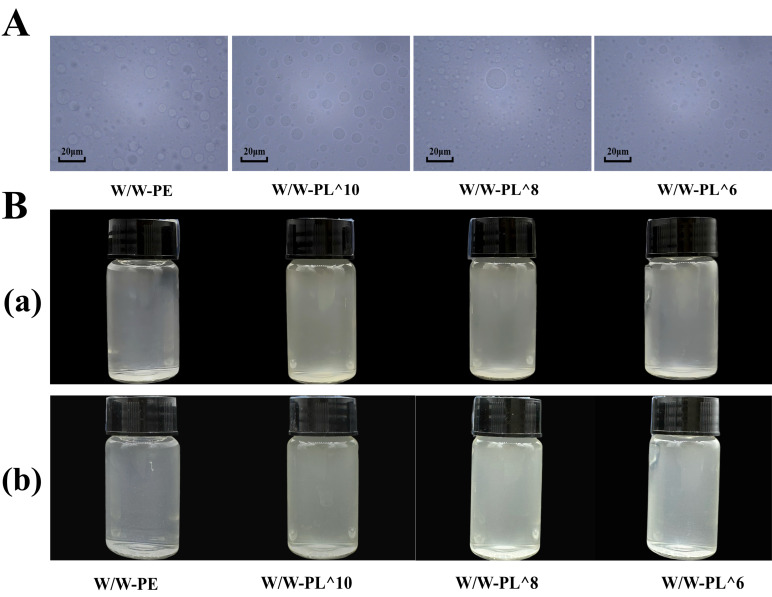
The morphology and storage state of Pickering emulsion: (**A**) microscopic morphology of fresh emulsion; (**B**) the appearance of Pickering emulsion on Day 0 (**a**); the appearance of Pickering emulsion on Day 15 (**b**); Note: the bottle used to seal the Pickering emulsion has a volume of 10 mL.

**Figure 3 foods-14-02729-f003:**
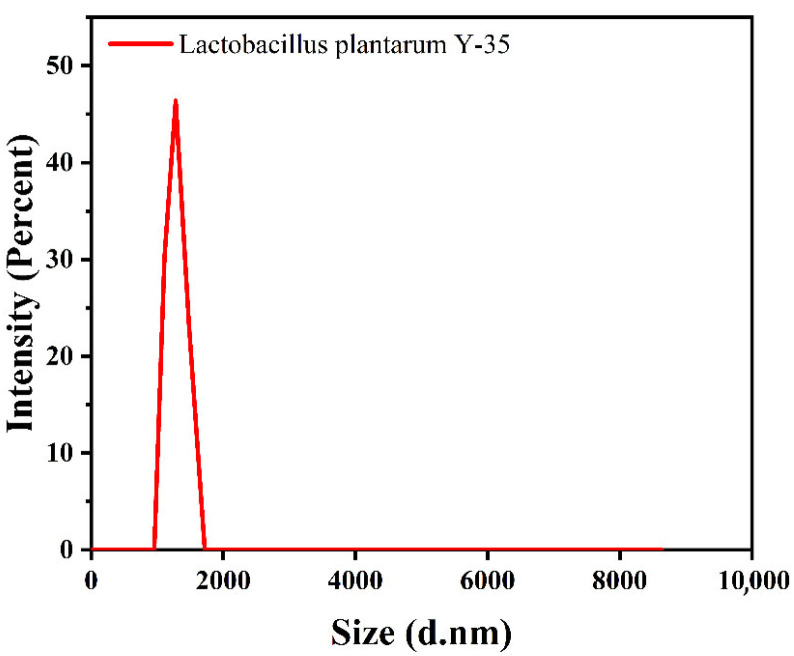
Particle size of *Lactobacillus plantarum*.

**Figure 4 foods-14-02729-f004:**
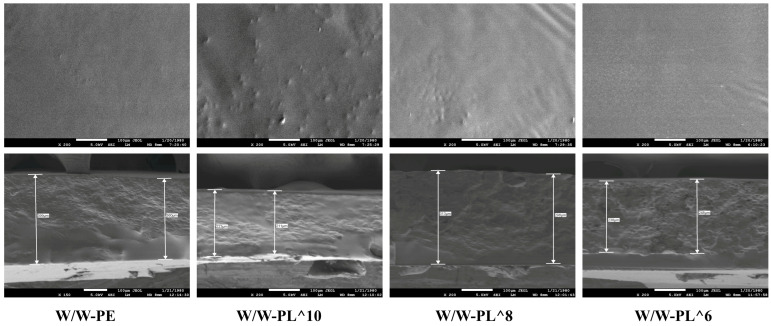
SEM image of the film: planar view of the film (×200) and cross-sectional view of the film (×100).

**Figure 5 foods-14-02729-f005:**
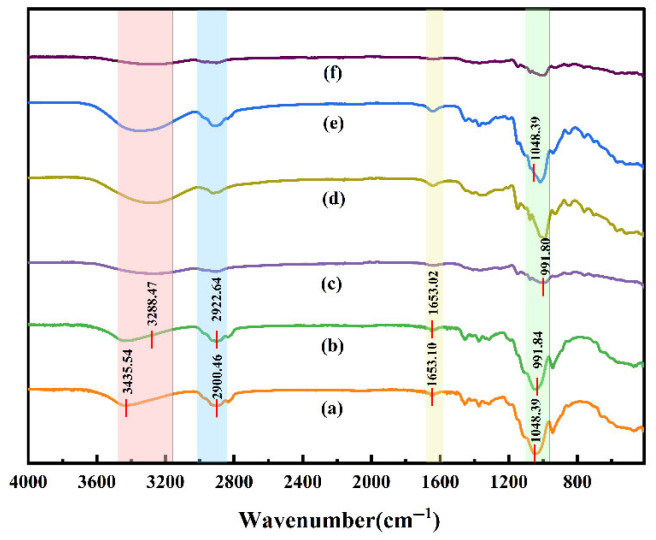
FTIR spectra of thin films (A): HPMC (**a**), MD (**b**), W/W-PE (**c**), W/W-PL^10 (**d**), W/W-PL^8 (**e**), W/W-PL^6 (**f**).

**Figure 6 foods-14-02729-f006:**
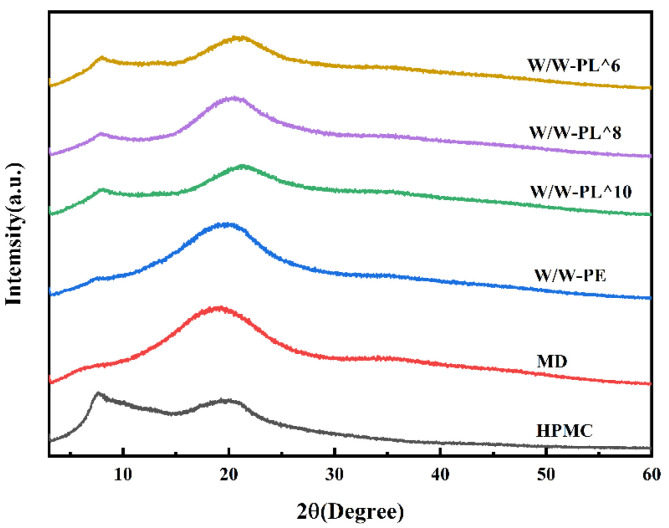
The XRD spectrum of the film.

**Figure 7 foods-14-02729-f007:**
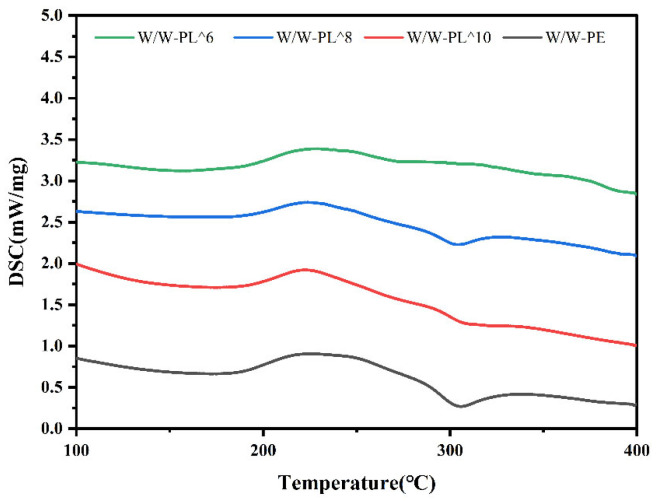
The DSC thermogram of the film.

**Figure 13 foods-14-02729-f013:**
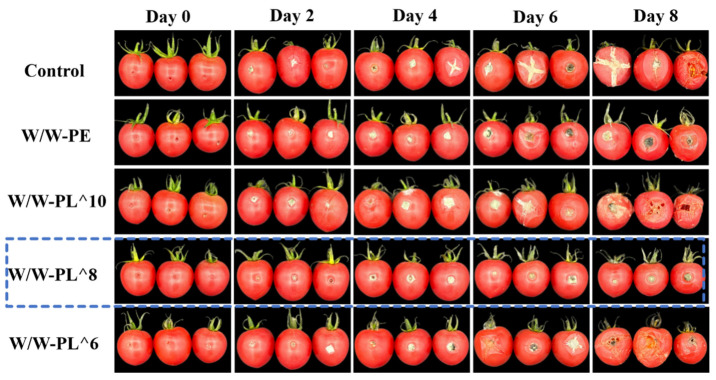
Preservation effect of cherry tomatoes after inoculation with *Alternaria alternata* under storage conditions of 25 ± 1 °C and 80% RH for 8 days.

**Figure 14 foods-14-02729-f014:**
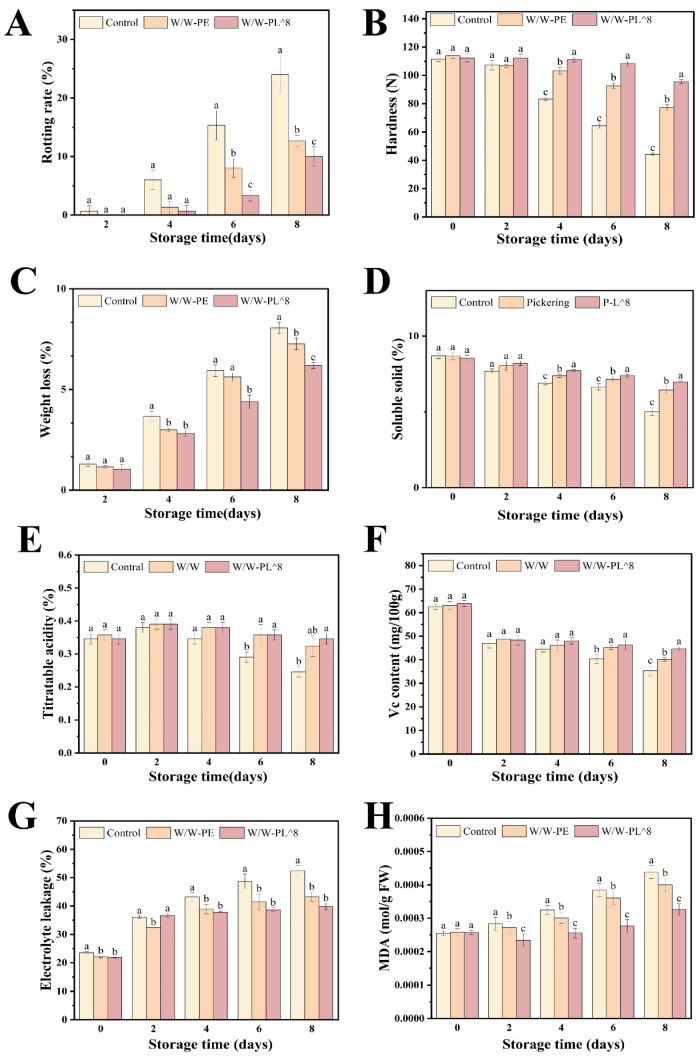
Postharvest indicators of cherry tomatoes: (**A**) fruit decay rate; (**B**) fruit hardness; (**C**) fruit weight loss rate; (**D**) total soluble solids content (SSC) of fruit; (**E**) titratable acidity (TA) of fruit; (**F**) vitamin C (VC) content of fruit; (**G**) electrolyte leakage of fruit; (**H**) malondialdehyde (MDA) content of fruit. Means in each cepa followed by different lowercase letters indicate significant differences (*p* < 0.05) according to Duncan’s test.

**Table 1 foods-14-02729-t001:** Particle size, zeta potential, and polydispersity index of Pickering emulsion and *Lactobacillus plantarum*.

Pickering Emulsions	Size (nm)	Zeta-Potential (mV)
W/W-PE	228.70 ± 63.66 ^b^	−16.4 ± 2.82 ^b^
W/W-PL^10	354.97 ± 88.24 ^ab^	−17.1 ± 2.45 ^b^
W/W-PL^8	393.03 ± 42.23 ^a^	−27.6 ± 3.27 ^c^
W/W-PL^6	290.70 ± 62.75 ^ab^	−13.9 ± 1.02 ^a^

The data are expressed as mean ± standard deviation. Means in each cepa followed by different lowercase letters indicate significant differences (*p* < 0.05) according to Duncan’s test.

## Data Availability

The original contributions presented in the study are included in the article, further inquiries can be directed to the corresponding authors.

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
