# Peer review of "Nature Nano-Barrier: HPMC/MD-Based Lactobacillus plantarum Pickering Emulsion to Extend Cherry Tomato Shelf Life"

_foods, 2025, doi:10.3390/foods14152729_

Round 1
Reviewer 1 Report
Comments and Suggestions for Authors
The manuscript entitled “Nature Nano-Barrier: Lactobacillus plantarum-Stabilized Pickering Emulsion as a Green Strategy for Extending Cherry Tomato Shelf Life” presents an interesting and high-quality piece of research. It holds significant potential for advancing sustainable approaches to maintaining the quality of perishable products, thereby contributing to shelf life extension and food loss reduction. However, some important details should be addressed to strengthen the manuscript.
The manuscript is not in the format required by the journal.
The text does not have numered lines which makes precise reference to comments difficult.
Names of Bacteria and molds should be in italics. Revise the title, abstract, and the whole manuscript.
The Latin name for tomato should be in italics. Revise the entire manuscript.
The following reference in the text should be revised: Laura et al. Laura is a name not a surname.
Revise the formula number 3. If the author is to subtract the weight of the empty container (w0) from the weight of the samples (w1), this must also be done in the numerator part of the fraction.
Some parts of the manuscript are in italics, but do not make sense (e.g. the first paragraph of the results and discussion section or the paragraph on statistical analysis).
The text is not aligned in some parts of the manuscript.
In the section 3.3.1., the first sentence should be located in materials and methods section.
Relative humidity should be abbreviated the first time it appears in the text (page 6). The same for water vapor transmission. After that, the authors shoud use the abbreviation. Revise the whole manuscript with all the used abbreviations.
In the first sentence of section 3.4.1., change ‘sensory evaluation’ to ‘visual evaluation’.
Reviewer 2 Report
Comments and Suggestions for Authors
Overall comments for the manuscript
- Correct the writing of microbial names by italicizing them, and follow the convention of abbreviating the microbial name after its initial full mention.
- Please adjust the font type and size throughout the entire manuscript to meet MDPI's specifications (Palatino). Title
Title
The title does not indicate which specific polysaccharide was utilized in the preparation of the Pickering emulsion. The title should be revised to encompass the full scope of the work.
Abstract
- No research objectives provided.
- The main pathogenic fungus, Alternaria, is not mentioned.
- For each result presented, numerical values should be added.
- Arrange the keywords in alphabetical order.
Introduction
- Restructure the content to ensure there's no repetition between paragraphs.
- First paragraph: The authors should consider stating or implying that cherry tomatoes are a key commercial or export fruit in China to boost the content's significance.
- First paragraph: What type of Alternaria alternata is it? Please provide more details.
- Second paragraph: What was the rationale for the authors' use of two polysaccharide types in the biomaterial packaging? Please explain the synergistic or interdependent relationship between them as the bio-packaging.
- Third paragraph: To maintain consistency, the authors need to clarify the relationship between LAB and Alternaria alternata, especially since Alternaria alternata was introduced early on.
- Fourth paragraph: To enhance clarity, the authors should include this information within the MD and HPMC paragraph, given their interconnectedness.
- Fourth paragraph: Identify the research gap or problem addressed by this work and integrate it into the relevant paragraph, along with the objectives.
Materials and Methods
- Correct the superscripting of microbial names; do not use the ^ symbol.
- Add references for every experiment.
- What does FW stand for in Equation 7?
Results and Discussion
The authors have not included a discussion of their experimental results; there are only references to other researchers' work. This section is crucial for demonstrating the new knowledge generated and explaining why the results of each experiment turned out as they did. This section must be revised significantly, as it is the core of the research. Additionally, the presentation of results should highlight data relationships, for example, by using correlation statistics. This would make the findings much more compelling.
Conclusion
Please include a section discussing unexplored avenues and future work, concluding with the practical applications of your research. Ensure this discussion aligns with your research objectives and addresses the research gap identified in the introduction.
Reviewer 3 Report
Comments and Suggestions for Authors
Dear Authors,
I have read your paper titled “Nature Nano-Barrier: Lactobacillus plantarum-Stabilized Pickering Emulsion as a Green Strategy for Extending Cherry Tomato Shelf Life” to Foods.
Your study presents an interesting aproach in using Lactobacillus plantarum-stabilized water-in-water Pickering emulsions for cherry tomato coating. However, I believe that the manuscript requires major revisions before it can be published.
Please note that all specific comments and suggestions have been incorporated directly into the manuscript as in-text comments for clarity and ease of reference.
I encourage you to revise your manuscript. I hope these suggestions will help strengthen your work and improve its scientific contribution.

Reviewer 4 Report
Comments and Suggestions for Authors
I consider this a manuscript with very interesting results that suggest an efficient alternative for maintaining the postharvest quality of cherry tomatoes. The study presents both the characterization of the emulsion and its effects on the physiology of cherry tomatoes, making it a comprehensive study. The manuscript is well written and well presented. In the PDF, I have included my detailed suggestions, which mainly involve correcting some formatting aspects, expanding the figure captions, and including discussion for some variables where only the results are currently presented.

Round 2
Reviewer 3 Report
Comments and Suggestions for Authors
Dear Authors,
I appreciate the effort made to address the previous comments and revise the manuscript: “Nature Nano-Barrier: HPMC/MD-Based Lactobacillus plantarum Pickering Emulsion to Extend Cherry Tomato Shelf Life” accordingly. However, I believe that further improvements are necessary to enhance the scientific clarity, consistency, and reproducibility of the study.
Please refer to the embedded comments and suggestions in the main text of the PDF, where I have indicated specific areas that require additional clarification, or methodological detail.
I encourage you to revise the manuscript thoroughly in line with these suggestions to strengthen the overall quality and impact of your work.
Wishing you all the best.
